# Extracellular Vesicles of the Plant Pathogen *Botrytis cinerea*

**DOI:** 10.3390/jof9040495

**Published:** 2023-04-20

**Authors:** Amelie De Vallée, Jean-William Dupuy, Christine Moriscot, Benoit Gallet, Solène Vanderperre, Gaëtan Guignard, Christine Rascle, Glen Calvar, Bastien Malbert, François-Xavier Gillet, Cindy Dieryckx, Mathias Choquer, Vincent Girard, Nathalie Poussereau, Christophe Bruel

**Affiliations:** 1Univ. Lyon, UCBL, INSA Lyon, CNRS, MAP, UMR5240, 69622 Villeurbanne, France; 2Plateforme Protéome, Univ. Bordeaux, 33000 Bordeaux, France; 3Univ. Grenoble-Alpes, CNRS, CEA, EMBL, ISBG, F-38000 Grenoble, France; 4Univ. Grenoble Alpes, CEA, CNRS, IBS, F-38000 Grenoble, France; 5Centre Technologique des Microstructures, Université Lyon 1, 69622 Villeurbanne, France; 6Univ. Lyon, Université Lyon 1, CNRS, ENTPE, UMR 5023 LEHNA, 69622 Villeurbanne, France; 7Bayer SAS, Crop Science Division Centre de Recherche La Dargoire, 69009 Lyon, France

**Keywords:** EV, fungus, secretion, proteomics, cell–cell communication, tomography, electron/confocal microscopy

## Abstract

Fungal secretomes are known to contain a multitude of components involved in nutrition, cell growth or biotic interactions. Recently, extra-cellular vesicles have been identified in a few fungal species. Here, we used a multidisciplinary approach to identify and characterize extracellular vesicles produced by the plant necrotroph *Botrytis cinerea*. Transmission electron microscopy of infectious hyphae and hyphae grown in vitro revealed extracellular vesicles of various sizes and densities. Electron tomography showed the co-existence of ovoid and tubular vesicles and pointed to their release via the fusion of multi-vesicular bodies with the cell plasma membrane. The isolation of these vesicles and exploration of their protein content using mass spectrometry led to the identification of soluble and membrane proteins involved in transport, metabolism, cell wall synthesis and remodeling, proteostasis, oxidoreduction and traffic. Confocal microscopy highlighted the capacity of fluorescently labeled vesicles to target cells of *B. cinerea*, cells of the fungus *Fusarium graminearum*, and onion epidermal cells but not yeast cells. In addition, a specific positive effect of these vesicles on the growth of *B. cinerea* was quantified. Altogether, this study broadens our view on the secretion capacity of *B. cinerea* and its cell-to-cell communication.

## 1. Introduction

Filamentous fungi are the most prevalent plant pathogens, causing dramatic reductions in the yield and quality of agricultural and horticultural production annually [1]. During the infectious process, these pathogens rely on the secretion of a complex repertoire of molecules, including toxins, plant-cell-wall-degrading enzymes and/or effector proteins. This secretion can reduce host defense and support the growth of the pathogen, allowing colonization of the host tissues. Several studies have described diverse fungal secretomes, revealing the compositions of their proteins and metabolites, their importance in the virulence of pathogens, and their variation in response to environmental modifications [2,3,4]. Recently, the presence of extracellular vesicles (EVs) in these secretomes has been investigated; however, their identification and molecular characterization in phytophatogenic fungi are still in their infancy.

EVs are released by bacterial, fungal, plant and animal cells [5,6,7]. They can carry proteins, nucleic acids (DNA, RNA), lipids, carbohydrates and metabolites. They represent a means of unconventional secretion that explains the observed and historically puzzling presence outside the cells of proteins of intracellular origin, such as metabolic enzymes [8,9]. EVs are also reported to play a role in inter-cellular communication between neighboring cells (inside a multi-cellular organism), or with non-self cells during host–pathogen interactions has also been reported [7,10,11,12,13].

EVs constitute a heterogeneous population of vesicles that can be sorted in three classes: exosomes (30–150 nm), micro-vesicles/ectosomes (100–1000 nm) and apoptotic micro-vesicles (800–1000 µm) or bodies (>1 µm) [14,15,16,17]. However, the isolation and distinction of these three classes are difficult because of the overlapping dimensions, shapes, densities, and compositions of EVs [16]. Moreover, when cellular origin cannot be documented using microscopy, a classification in small (<100–200 nm) and large EVs (>200 nm) is recommended [18,19].

The biogenesis of EVs is still unclear at the molecular level. It has been proposed that micro-vesicles shed from the plasma membrane or originate from inverted macropinocytosis, while exosomes originate as intra-luminal vesicles inside the endosome (during the maturation of the latter into multi-vesicular bodies (MVB)) and are released upon fusion of the MVB with the plasma membrane [17,20,21,22]. In all cases, membrane curvature would be required, relying on changes in lipid composition/distribution, on the recruitment of the endosomal sorting complex required for transport machinery (ESCRT) and/or on tetraspanins [23,24,25,26,27,28]. Studies on *Saccharomyces cerevisiae* and the yeast human pathogen *Cryptococcus neoformans* also suggest that exophagy (fusion of autophagosome with the plasma membrane) and components of the conventional and non-conventional secretory pathways of cells are also involved [16,29,30,31,32].

In fungi, EVs were first observed in the early 1970s using transmission electron microscopy (TEM) at the surface of protoplasts of the fungus *Aspergillus nidulans* [33] and at the surface of cells of *C. neoformans* [34]. In parallel, multi-vesicular bodies (MVB)-like structures were observed in fungal cells [35], and later in yeast protoplasts [36]. Together with TEM images of vesicles located between the fungal plasma membrane and the cell wall [37], these observations paved the way for the first characterization of fungal EVs in the yeast *C. neoformans* in 2007 [38]. EVs were then isolated in 2014 from the filamentous fungus *Alternaria infectoria*, a ubiquitous environmental fungus that both spoils food crops and causes human infections [39]. They have since been identified in more than 20 yeasts and filamentous species [40,41,42,43].

Mostly investigated in human pathogenic species and in *S. cerevisiae* [22,31,44,45,46,47,48,49,50,51,52], the protein composition of fungal EVs is shown to be diverse, including proteins involved in cell division, transport, signaling, metabolism (carbohydrate, lipid, nucleotide), oxidoreduction, stress response, protein biosynthesis/degradation, cell wall remodeling and virulence [41,43,53,54]. In addition, polysaccharides [38,50], pigments [41,43] and a variety of RNA species have been found in some EVs [43,55]. In the field of fungal-related plant pathology, research has begun on the plant side, suggesting that EVs play a role in plant defense against fungi [11,56,57,58]. On the fungal side, EVs were identified in *Fusarium oxysporum* and *F. graminearum*, carrying proteins and effectors involved in virulence [59,60]. EVs were also identified in *Colletotrichum higginsianum*, the causal agent of anthracnose disease [61].

*B. cinerea* is an ascomycetous fungus responsible for gray mold disease. It can infect hundreds of dicotyledonous plants and was listed among the ten most devastating plant pathogenic fungi worldwide [62]. The necrotrophic strategy of this pathogen relies on the secretion of virulence factors [3] that kill plant cells prior to their invasion and degradation. Hence, characterization of the secretome of *B. cinerea* is important to understanding its pathogeny. This study explores the production of EVs by *B. cinerea* and their potential roles in its biology.

## 2. Materials and Methods

### 2.1. Strains and Culture Conditions

All strains were grown at 21 °C. Strain B05.10 of *B. cinerea* (teleomorph *Botryotinia fuckeliana* (de Bary) Whetzel) was maintained on sporulation medium (MS) [63] and grown in MS, potato dextrose agar (PDB) or PDB diluted by half (½ PDB) or by a quarter (¼ PDB). *F. graminearum* was maintained in 25% V8 vegetable juice, 3.75 g/L CaCO_3_, and 25 g/L agar. For TEM ultrastructure of *B. cinerea* grown in solid culture: mycelium explants were used to inoculate MS plates overlaid with cellophane sheets and incubated for 3 days at 21 °C in the dark. The cellophane was then transferred to a clean medium-free Petri dish for fixation. For TEM ultrastructure of *B. cinerea* grown in liquid culture: mycelium explants were used to inoculate ½ PDB or conidia (10^5^/mL) were used to inoculate a minimum medium (2 g/L NaNO_3_, 1 g/L K_2_HPO_4_, 0.5 g/L KCl, 0.5 g/L MgSO_4_·7H_2_O, 0.01 g/L FeSO_4_·7H_2_O and 2% (*w*/*v*) glucose). Cultures were agitated for 4 days at 110 rpm, with medium renewal on day 3. The mycelia were filtered (100 µm) prior to fixation.

### 2.2. Preparation of EVs

For EVs production, conidia were collected from 11-day-old cultures on MS and used to inoculate (2 × 10^5^ conidia per plate) the surface of cellophane sheets overlaying solid MS [2]. Fifty plates per experiment were prepared and incubated for 3 days at 21 °C in the dark.

All steps were performed at 4 °C. The mycelium was harvested and gently rotated for 4 h in 300 mL of cold PBS (phosphate–buffer saline). Following filtration (200 µm), cell debris were removed from the filtrate by two successive centrifugations (20,000 g; 20 min). The resulting supernatant was filtered twice (40 µm and 1.2 µm) and centrifuged for 1 h at 100,000 g (Beckman rotor 30.50 Ti, Beckman Coulter, Brea, CA, USA). The pellet containing the vesicles was solubilized in 2 mL of cold PBS. OptiPrep™ (Serumwerk Bernburg, Bernburg, Germany) 60% *w*/*v* iodixanol was diluted to 40, 20, 10 and 5% in 12.25% (*w*/*v*) sorbitol, 1 mM EDTA and 10 mM Tris-HCl pH 7.4, and a discontinuous gradient (5–40%) was formed by layering 3 mL of each solution in polypropylene centrifugation tubes. The vesicles were loaded onto this gradient and centrifuged at 100,000 g for 16 h (Beckman rotor JS 24.15 swing, Beckman Coulter). Fractions (2 mL) were collected from the top of the gradient and their density was determined using spectrophotometry (244 nm) according to the manufacturer’s instructions (Serumwerk Bernburg, Bernburg, Germany). Protein concentrations were obtained using Qubit^TM^ 2 fluorimeter (Invitrogen, Willow, TX, USA) and the Qubit^TM^ protein assay kit according to the manufacturer’s instructions, or with BCA protein assay (Thermo Scientific™, Rockford, IL, USA) for CFSE-BcEVs. The fraction of interest (fraction 5) was washed twice in PBS (100,000 g; 1 h), suspended in 300 µL of cold PBS, and stored on ice before use, or it was placed at −80 °C for conservation.

### 2.3. EVs Labelling with CFSE

The 100,000 g pellet was suspended in 2 mL of cold PBS. Carboxyfluorescein succinimidyl ester (CFSE, ebioscience, Carlsbad, CA, USA) was added (40 µM) and the sample was incubated for 1h in the dark at 21 °C before loading onto the OptiPrep™ gradient. The fluorescence in the recovered fractions was measured using a TECAN infinite M1000 (excitation, 494 nm; emission, 521 nm). In parallel, PBS (2 mL) containing 40 µM CFSE (no biological material) was subjected to the same protocol, and fraction 5 was recovered to be used as PBS control sample.

### 2.4. Dynamic Light Scattering

Particle size distribution of EVs preparations was determined by dynamic light scattering (DLS) at 19 °C using a Zetasizer nano series (Malvern Instruments, Malvern, UK). Eight biological replicates and three technical replicates (ten reads each) were performed.

### 2.5. Microscopy

*Sample Preparation—*Mycelia were fixed in 4% paraformaldehyde (PFA) and 0.4% glutaraldehyde (GA) in 0.1 M PHEM (PIPES, HEPES, EGTA, MgCl_2_ buffer pH 6.9) for 30 min at room temperature (RT) with gentle shaking. The buffer was discarded, and a second incubation (30 °C, RT) was performed in 0.1 M PHEM buffer containing 2% PFA and 0.2% GA. High-pressure freezing was performed as described in [64]. Briefly, the mycelium was deposited on the 200 μm side of a 3 mm type A gold plate (Leica Microsystems, Wetzlar, Germany) covered with the flat side of a 3 mm type B aluminum plate (Leica Microsystems, Wetzlar, Germany) and vitrified using an HPM100 system (Leica Microsystems, Wetzlar, Germany). After freezing, the samples were cryo-substituted in an AFS2 machine (Leica Microsystems, Wetzlar, Germany) as follows: after 40 h at −90 °C in acetone with 1% OsO_4_, the samples were slowly warmed to −60 °C (2 °C/h), stored for 10 h at −60 °C. Then, the temperature was raised to −30 °C (2 °C/h), the samples were stored for a further 10 h at −30 °C before a 1 h incubation at 0 °C, and storage at −30 °C until further processing (all these steps are automatically performed in the AFS2 instrument using a pre-defined program). Then, the samples were rinsed 4 times in pure acetone before being infiltrated with progressively increasing concentrations of resin (Epoxy embedding medium, Sigma, st Quentin Falavier, France) in acetone while increasing the temperature to 20 °C. At the end, pure resin was added at RT and the samples were placed at 60 °C for 2 days of polymerization.

*Transmission Electron Microscopy (TEM) of fungal cells grown* in vitro—70 nm thin sections were cut from high-pressure-frozen-resin-embedded samples, using a UC7 ultramicrotome (Leica Microsystems, Wetzlar, Germany) and collected on 100 mesh Formvar carbon copper grids. The sections were post-stained with 2% uranyl acetate and lead citrate (5 min each). Samples were observed using a Tecnai G2 Spirit BioTwin microscope (FEI company, Hillsboro, OR, USA) operating at 120 kV with an Orius SC1000B CCD camera (Gatan, Pleasanton, CA, USA).

*Electron tomography*—For electron tomography, 200–350 nm thick sections were used. Tomogram tilt series were collected at ×6500 from −40° to +40° or ×9600 from −50° to +50° with 2° intervals on a Tecnai TF20 microscope operating at 200 kV (FEI company, Hillsboro, OR, USA) and using a CCD (4 k × 4 k) camera (Gatan, Pleasanton, CA, USA). Tomography data acquisition, automated centering, focus adjustment, tilt setting, and image capture were performed using the SerialEM software 3.8.0 beta [65]. Typical size pixel at the sample level was 15.92 nm (×6500) or 10.68 nm (×9600) with a defocus target set to −5 microns. Tomograms were reconstructed using IMOD software 4.11.11 [65,66] while contours of cellular features were modelled using the 3dmod software 4.11.11.

*TEM of fungal cells grown in planta—*Infection assays were performed using primary French bean leaves (*Phaseolus vulgaris* var Saxa) inoculated with conidia (10^4^/mL in ¼ PDB) and incubated 40 h at 21 °C under 80% relative humidity and dark–light (16 h/8 h) conditions. Infection zones were collected and fixed in 0.1 M phosphate buffer, pH 6.8, containing 2.5% GA and 4% PFA (2 h at RT including the first 30 min under vacuum, and then 4 h at 4 °C). Following 6 washes in 0.2 M phosphate buffer, pH 6.8 (5× 30 min and overnight; 4 °C), 2 mm leaves band were cut out, treated with 2% osmium in 0.1 M phosphate buffer, pH 6.8 (2 h; 4 °C), rinsed in water (3× 30 min) and dehydrated through incubations in 30, 50, 70, 80, 90, 95% ethanol baths (2× 15 min each) and in 100% ethanol (3× 10 min). The samples were then impregnated through three successive Ethanol/Spurr’s resin solutions (ratio (*v*/*v*) 3/1 (2 h), 1/1 (2 h) and 1/3 (3 h)). The leave bands were heated to 70 °C for 3 days in Spurr’s resin for polymerization on silicon molds. Ultra-thin sections were cut and observed onto TEM–copper–Formvar grids (75 meshs) with a TEM Philips CM120 operating at 120 kV.

*TEM of isolated EVs—*Five μL of EVs suspension in PBS were deposited onto a TEM–copper–Formvar grid and left decanted for 30 s. Excess liquid was absorbed using Whatman paper and the grid was flipped over a drop of 2% uranyl acetate (UA) for 30 s. UA was removed and the grid was washed with distilled water. The grid was then placed into the chamber of a transmission electron microscope JEOL 1400 Flash.

*Confocal microscopy*—Observations were performed in 96-well micro-plates (Greiner Bio-one, Frickenhausen, Germany). Conidia of *B. cinerea* or *F. graminearum* were suspended in 200 µL (5 × 10^3^/mL) of ¼ PDB and incubated at 21 °C in the dark, for 6 h, 10 h or 14 h. After removal of 160 µL of medium, 30 µL of CFSE-BcEVs (Fraction F5) in PBS buffer were added, and the plate was agitated (300 rpm) at 21 °C in the dark for 1–4 h. The control experiments used 30 µL of the PBS-CFSE (Fraction 5). When used, 20 µL of *S. cerevisiae* cells (2.5 × 10^5^/mL), grown in ¼ PDB at 28 °C and collected during the exponential phase, were added. Aniline blue (0.005% final) was directly added to the samples from a 10× solution in water. For onion epidermis observations, 1.5 cm^2^ of epidermis was deposited onto 10 µL of CFSE-BcEVs (fraction F5) on a microscope slide and incubated in the dark for 3 h at 21 °C. Before observation, the onion epidermis was rinsed with PBS and mounted with a coverslip. All observations were performed with a Zeiss axio Observer 7 LSM-800 confocal microscope (Zen 2.6 Blue software) set at 493 nm (excitation) and 817 nm (emission) for CFSE detection and at 353/465 nm for aniline blue detection.

### 2.6. Fungal Growth in the Presence of EVs

Conidia of *B. cinerea* or *F. graminearum* were suspended in 100 µL PDB (5 × 10^3^/mL) in 96-well micro-plates, and incubated at 21 °C for 2 h in the dark. After the removal of 75 µL of medium, 50 µL of sterile water and 25 µL of BcEVs (Fraction 5) representing 5.5 ± 1.8 µg of total protein (BCA protein assay) were added. The plate was incubated at 21 °C and growth was measured over time at 620 nm using a TECAN infinite F200Pro.

### 2.7. Label-Free Quantitative Proteomics

Extracellular vesicles preparations were concentrated to 40 µL on Vivaspin 500 (3000 MWCO PES, Sartorius, Stonehouse, UK), solubilized in Laemmli sample buffer (Biorad, Hercules, USA) and boiled (100 °C) for 3 min. Samples with an equal protein concentration (10 µg) were loaded onto a 10% acrylamide SDS-PAGE and the migration was stopped when the proteins entered the resolving gel. The steps of sample preparation and in gel digestion by trypsin were performed as previously described [67]. NanoLC-MS/MS analysis were run using an Ultimate 3000 RSLC Nano-UPHLC system (Thermo Scientific, Carlsbad, CA, USA) coupled to a nanospray Orbitrap Fusion™ Lumos™ Tribrid™ Mass Spectrometer (Thermo Fisher Scientific, Carlsbad, CA, USA). Each peptide extracts were loaded onto a 300 µm ID × 5 mm PepMap C18 precolumn (Thermo Scientific, Carlsbad, CA, USA) at a flow rate of 10 µL/min. After a 3 min desalting step, peptides were separated on a 50 cm EasySpray column (75 µm ID, 2 µm C18 beads, 100 Å pore size, ES903, Thermo Fisher Scientific, Carlsbad, CA, USA) with a 4–40% linear gradient of 0.1% formic acid in 80% ACN in 58 min. The separation flow rate was set at 300 nL/min. The mass spectrometer operated in positive ion mode at a 1.8 kV needle voltage. Data were acquired using Xcalibur 4.1 software in a data-dependent mode. MS scans (*m*/*z* 375–1500) were recorded at a resolution of R = 120,000 (@ *m*/*z* 200) and an AGC target of 4 × 105 ions collected within 50 ms, followed by a top speed duty cycle of up to 3 s for MS/MS acquisition. Precursor ions (2 to 7 charge states) were isolated in the quadrupole with a mass window of 1.6 Th and fragmented with HCD@30% normalized collision energy. MS/MS data were acquired in the ion trap with rapid scan mode, AGC target of 3 × 103 ions and a maximum injection time of 35 ms. Selected precursors were excluded for 60 s. Protein identification and label-free quantification (LFQ) were carried out using Proteome Discoverer 2.4. MS Amanda 2.0, Sequest HT, and Mascot 2.5 algorithms were used for protein identification in batch mode by searching against a ENSEMBL *B. cinerea* ASL83294v1 database (13,023 entries, release 47). Two missed enzyme cleavages were allowed for the trypsin digestion. Mass tolerances in MS and MS/MS were set to 10 ppm and 0.6 Da. Oxidation (M) and acetylation (K) were searched as dynamic modifications and carbamidomethylation © as a static modification. Peptide validation was performed using Percolator algorithm [68] and only “high confidence” peptides were retained corresponding to a 1% false discovery rate at peptide level. Minora feature detector node (LFQ) was used along with the feature mapper and precursor ions quantifier. The quantification parameters were selected as follows: (1) Minimum of 2 unique peptides, (2) precursor abundance based on intensity, (3) normalization mode: total peptide amount, (4) protein abundance calculation: summed abundances, (5) protein ratio calculation: pairwise ratio based, (6) imputation mode: none, and (7) hypothesis test: t-test (background-based; *p*-value < 0.05). The mass spectrometry proteomics data were deposited in the ProteomeXchange Consortium via the PRIDE [69] partner repository with the dataset identifier PXD035766.

### 2.8. Data Analysis

Functional enrichment Gene Ontology (GO) analyses were performed using gProfiler [70,71]. Putative transmembrane domains were predicted using TMHMM v2.0 [72]. Putative signal peptides without a transmembrane domain were predicted with SignalP-5.0 [73]. The prediction of effectors and apoplast-targeted proteins were performed using the EffectorP and ApoplastP tools [74,75]. The prediction of cellular localization of the proteins was performed using the DeepLoc-1.0 tool [76]. The classification of CAZy was extracted from the Carbohydrate Active Enzymes database [77]. Additional prediction tools and databases were KEGG, MEROPS, and InterPro, as well as manually curated *B. cinerea* protein lists, including plant-cell-wall-degrading enzymes (PCWDE), fungal cell wall enzymes (FCWE), transporters, purine metabolism, virulence factors, proteases, laccases [2,78,79]. The prediction of protein–protein interactions was performed using STRING [80].

## 3. Results

### 3.1. Production of Extracellular Vesicles (EVs) by B. cinerea

The capacity of *B. cinerea* to produce EVs was first investigated using transmission electron microscopy (TEM). To this end, the fungus was grown in liquid medium using conidia or mycelial explants as an inoculum. In parallel, the fungus was also grown on solid medium overlaid with cellophane. In hyphae collected from all cultures, vesicles could be detected using TEM between the fungal plasma membrane and the cell wall. Groups of vesicles heterogeneous in size (50–500 nm) and in electron densities were detected (Figure 1A,B and Appendix A). These vesicles resembled those contained in MVB identified inside the hyphae (Figure 1B and Appendix A) and images of apparent pre- and post-fusion events between the plasma membrane and MVB were captured that suggested a likely MVB origin for these groups of EVs (Figure 1C and Appendix A). A smaller population of individual vesicles, of unclear origin, was rather homogeneous in size (80–100 nm) and exhibited a weak electron density (Figure 1A and Appendix A). To obtain more detailed structural and three-dimensional information, electron tomography was applied to the samples. The deformation of MVB towards the adjacent plasma membrane and MVB connected to this membrane were observed (Figure 1 D and Appendix A). In addition, tomography images revealed a diversity in vesicle shapes, from spherical to tubular (Figure 1D and Appendix A). Altogether, these results indicate that *B. cinerea* produces EVs of various sizes, shapes, and densities during vegetative growth in liquid and in solid cultures; these EVs are hereafter referred to as BcEVs. Whether the fungus also produced BcEVs during plant infection was then addressed using TEM images of infected bean leaves. As shown in Figure 1E, BcEVs, both individually and in groups, can be identified between the plasma membrane and the cell wall of hyphae localized inside or on the surfaces of plant cells.

### 3.2. Isolation of BcEVs

Having observed that BcEVs could be produced in liquid as well as on solid media, their isolation was attempted from both cultures. The fungus was first grown for 3 days in a liquid medium. The cultured supernatant was subjected to filtration and differential centrifugations in order to eliminate cell debris and to ultimately collect particles, sedimenting at 100,000 g and containing spheroid vesicles (30–400 nm), as shown by TEM. Unfortunately, a gelatinous material that co-sedimented with these particles hampered their manipulation too much for this protocol to be adopted.

*B. cinerea* was then grown on cellophane sheets overlaying solid media and the secreted material was recovered via gentle agitation of the mycelium detached from the cellophane in PBS buffer. Filtration, differential centrifugations and additional separation through an iodixanol density gradient allowed one fraction out of six (F5, Figure 2A) to be identified. This showed the highest protein content and a density (1.142) similar to that described in the literature for the purification of EVs [81,82]. When FM4-64, a lipophilic molecule used to dye membranes, was added to the sample before separation through the density gradient, fraction F5 concentrated most (91.7%) of the fluorescent signal recovered in fractions 1 to 6 (F1 and F2, 1.2%; F3, 1.5%; F4, 2.7%, F6, 1.8%). Additionally, when CFSE, a membrane permeable dye commonly used to label the lumen of EVs [83,84], was used instead of FM4-64, fraction F5 was again the fraction concentrating most (80.7%) of the fluorescent signal (Figure 2A). The material contained in F5 formed a non-gelatinous pellet and TEM analysis demonstrated that it contained vesicles, 50 to 500 nm in size, similar to EVs described in the literature (Figure 2B). Dynamic light scattering analysis of this fraction further revealed particles with sizes concentrated in a range from 90 to 500 nm (peak at 190 nm; Figure 2C), in accordance with the TEM results shown in Figure 1 and Figure 2B. Considering that lipophilic dyes can aggregate and form micelles and/or alter the physical properties of EVs [84], replicates were prepared using CFSE and the reproducible fluorescence distribution as well as the reproducible amounts of proteins in each collected fraction (Figure 2A) established the isolation protocol as robust. Furthermore, these experiments validated the possibility to label BcEVs, while at the same time eliminating the excess of dye (fractioning in the top part of the density gradient).

### 3.3. Proteome Analysis of BcEVs

Mass spectrometry was used to characterize the proteome of EVs isolated from the *B. cinerea* cultures. BcEVs samples were prepared from four independent solid cultures and their protein contents were subjected to trypsin digestion and LC-MS/MS analysis. A total of 2461 proteins were identified with a minimum of two peptides in all four replicates. Only 8.8% of these proteins were predicted as signal peptides (SP)-containing proteins (Appendix A), indicating a good separation of the EVs from the free proteins secreted by the fungus via the conventional secretory pathway.

Due to biological and technical variability, the normalized abundance of the 2461 proteins present in the four replicates varied to different degrees (Appendix A). We considered that proteins associated with the EVs in a non-specific way were likely to show more abundance variation between replicates than EV-specific proteins. A maximum abundance variation of 1.5 across the four replicates was therefore used to filter potential unspecific EV proteins out of the proteomics data. This restricted the analysis to 673 proteins (Appendix A).

#### 3.3.1. Functional Classification of Trans-Membrane and GPI-Anchored Proteins in BcEVs

Predictions of transmembrane (TM) domains and glycosylphosphatidylinositol (GPI) anchors were used to list 271 proteins containing 1 to 16 putative TM and/or a GPI anchor, representing 40% of the 673 proteins of interest (Appendix A). Gene ontology analysis (GO, biological process (BP)) of these proteins revealed enriched categories, the top 10 of which (*p*-value < 6 × 10^−3^) highlighting localization, transport and beta-glucan biosynthesis (Figure 3A). Prediction of these proteins localization mainly highlighted the endoplasmic reticulum (ER), the mitochondria and the cell membrane (Figure 3B). Based on the enzyme commission (EC) number included in the KEGG sheets of these proteins, 49 of them were predicted to be oxido-reductases, transferases (of sugars, phosphorous, acyl or methyl groups), hydrolases (acting on proteins, sugars, lipids) or translocases (of phospholipids or calcium) (Figure 3C). Additional prediction tools and databases, as well as manually curated *B. cinerea* protein lists, enabled the classification of 206 of these 270 proteins (76%) into eight functional categories (Appendix A): transport, metabolism, proteostasis, oxidoreduction, fungal-cell-wall-related enzymes (FCWE), vesicular trafficking and signaling (Figure 3D). Sub-categorization allowed for a more detailed classification, revealing transporters of eight different classes, metabolic pathways linked to seven cellular requirements, components of the protein folding, modification and sorting machineries, or enzymes related to the biosynthesis of cell wall polysaccharides: chitin, glucan, and galactosaminogalactan. (Figure 3D).

#### 3.3.2. Functional Classification of Soluble Proteins in BcEVs

Among the 673 proteins of interest, 402 (60%) were predicted not to contain any TM domain or GPI anchor. As shown in Figure 4A, the top 10 enriched categories (*p*-value < 7.2 ×$10^{-7}$) revealed via a GO (BP) analysis of these proteins highlighted cellular localization, intracellular transport, electron transport, vesicle-mediated transport and amide biosynthesis. Prediction of these proteins localization mainly highlighted the cytoplasm (33%), the mitochondria (28%) and the nucleus (13%) (Figure 4B). The retrieval of the EC number from the KEGG sheets of 125 of these 402 proteins allowed them to be sorted into six classes of enzymes, namely transferases, hydrolases, oxidoreductases, translocases, ligases and lyases (Appendix A and Figure 4C). Finally, 351 of these 402 proteins could be classified into 12 functional categories (Appendix A and Figure 4D). Five categories containing at least twenty-five proteins could be sub-categorized. Thus, the category ‘metabolism’ includes energy, nitrogen, sugars, carbon, lipids, cofactors, sterols and secondary metabolism. The category ‘DNA/RNA metabolism’ includes replication, transcription, translation and RNA degradation. ‘Proteostasis’ includes the biogenesis, folding/assembly, sorting, modification and degradation of proteins, and ‘vesicular trafficking’ includes the sorting, coating, tethering, transport and fusion of vesicles. Lastly, ‘Intracellular transport’ groups together nuclear import/export, and carriers of metabolites or lipids/sterols. Five other categories, made of at least eight members, distinguish ribosomal and cytoskeleton proteins, as well as proteins involved in signaling, oxidoreduction and cellular processes. BcEVs therefore contain numerous cytosolic proteins deriving from various compartments of the fungal cell and involved in multiple biological processes. Only 17 of them contain a signal peptide. Investigation of putative interaction networks between the proteins of each sub-category was next investigated and revealed a high connectivity (60–100%) between the proteins classified in ‘metabolism’, ‘DNA/RNA metabolism’, ‘proteostasis’, ‘vesicular trafficking’, signaling and ‘intracellular transport’ (Appendix A).

Only one known virulence factor (BcSun1) and one cell-death-inducing protein (BcGs1) were found in the 673 BcEV protein of interest. No candidate effector protein could be identified in this dataset when the Effector-P prediction tool was used. However, BcGs1 is described as an effector of *B. cinerea* [85] and five proteins (BcB431, BcFlp2, BcGas1, Bcin03g00890, Bcin03g02710) belong to four sequence-unrelated structurally similar (SUSS) effector groups/clusters that contain at least one known effector identified in phytopathogenic fungi [86] (Appendix A). Finally, 30 proteins were predicted as apoplastic proteins using the Apoplast-P prediction tool, including BcGs1, BcSun1, BcGas1 and Bcin03g00890. Notably, 14 of these 30 proteins are predicted as membrane proteins (Appendix A).

### 3.4. Marker Proteins in BcEVs

Among the 673 proteins identified in BcEVs, 32 were predicted by KEGG as exosomal proteins in *B. cinerea* (Appendix A). These proteins include two prohibitins (BcPhb1 and BcPhb2), two heat shock proteins (Bcin10g00300 and Bcin10g00300), three actin binding proteins (BcSac6; BcArp2; BcArp3), two G-proteins (BcG1 and Bcin11g04620), two vesicular coat proteins (BcSec26 and Bcin02g05260) and three ESCRT proteins (BcVps4; BcDid2; BcSnf7). Moreover, 10 orthologs of the established EV markers in *Candida albicans* [87] can be found in the 673 BcEVs proteins, including the t-SNARE BcYkt6, the chitin synthase BcChsIV, the glucan synthase BcFks1, the glucanosyltransferase BcGas1 (involved in the elongation of glucan chains), the laccase BcLcc13, the pore membrane protein BcPom33 and the prohibitin BcPhb2 (Appendix A). In addition, the autophagy protein BcAtg8, the tetraspanin Tsp3 and several Sec proteins, including the exocyst protein BcSec6 belong to the 673 proteins identified in BcEVs. Such proteins are used as EV markers in mammalian systems [88,89].

### 3.5. Labelled BcEVs can Stain Plant Cells

Drops of fluorescently labeled BcEVs (CFSE-BcEVs) were deposited onto onion epidermis fragments. Onion epidermis is an established experimental model for evaluating plant penetration by *B. cinerea*. In addition, it lacks chloroplasts that emit a strong auto-fluorescence signal at the same wavelengths used to detect CFSE. After 3 h of contact, a CFSE signal could be detected at the surface of the plant cells using confocal microscopy (Figure 5A). A fluorescent signal was also detected intra-cellularly, surrounding the cell nucleus (Figure 5B).

Whether BcEVs could have an effect on a host plant of *B. cinerea* was then explored. First, BcEVs were deposited onto bean leaves. In comparison to the mock experiment, this triggered no visible reaction of the plant. To circumvent a possible limitation of BcEVs penetration through the plant cuticle, infiltration assays on tobacco leaves were then performed. The leaves showed no response to BcEVs, whereas the yellowing of their tissues was visible when the raw *B. cinerea* secretome was infiltrated as a positive control (Appendix A). In addition, leaf ROS production could not be detected in areas infiltrated with the BcEVs suspension, whereas high production was detected where the raw secretome was infiltrated (Appendix A).

### 3.6. Labelled BcEVs can Stain Fungal Cells

Conidia of *B. cinerea* having germinated in liquid medium were incubated with CFSE-BcEVs and then analyzed using confocal microscopy (Figure 6A). At one hour of incubation, a fluorescent signal could be monitored at the periphery of both conidia and germ tubes. At 4 h, the fluorescent signal could be detected as dots, or diffused inside the cytoplasm of some germ tubes. Noticeably, when yeast cells (*S. cerevisiae*) collected in exponential phase were added together with CFSE-BcEVs to the germinated conidia of *B. cinerea*, the periphery of the *B. cinerea* cells was stained within an hour, but that of the yeast cells was not (Figure 6B), and longer incubation did not modify this result. The glucans marker, aniline blue, stained the wall of all cells and its signal co-localized with that of CFSE, indicating that the BcEVs interact with the cell wall of *B. cinerea* and not with the cell wall of *S. cerevisiae* (Figure 6B). In a parallel experiment, conidia of both *F. graminearum* and *B. cinerea* were incubated in liquid medium. After germination, the fungal cells were mixed with CFSE-BcEVs and analyzed by confocal microscopy. After 1h of incubation, conidia and germ tubes of both fungi exhibited a similar fluorescent signal highlighting their surfaces (Figure 6C). As in Figure 6A, fluorescent dots were not visible inside *B. cinerea* cells at that time, while some were already visible inside cells of *F. graminearum*.

### 3.7. BcEVs Can Stimulate Cell Growth in B. cinerea

*B. cinerea* was grown in liquid medium supplemented with BcEVs. The fungal growth was stimulated by 22% in the presence of CFSE-BcEVs (Figure 7A) and confocal microscopy performed at the end of the experiment (55 h culture) showed fluorescent vacuoles inside the hyphae (Appendix A). Since BcEVs contain lipids and proteins, one possible explanation of the results was the additional nutrient source they may have represented (the BcEVs samples averaged 5.5 ± 1.8 µg total proteins). If this were the case, however, then the BcEVs should also stimulate the growth of other absorbotrophy-feeding filamentous fungi. Having shown that CFSE-BcEVs could stain cells of *F. graminearum* (Figure 6C), this fungus was selected to test the effect of BcEVs on its growth under the same conditions used for *B. cinerea*. As shown in Figure 7B, the growth curves recorded in the presence or absence of EVs were identical.

## 4. Discussion

During plant infection, phytopathogenic fungi secrete numerous molecules that participate in the molecular dialogue occurring at the interface between the two organisms [90]. In the case of *B. cinerea*, the agent that causes gray mold disease, this supports a necrotrophic strategy of host invasion [2,3]. It relies on the conventional secretory pathway to secrete SP-containing proteins such as lytic enzymes and cell-death-inducing proteins [91,92,93]. Unconventional secretion has not been investigated in this fungus and this study explored its capacity to produce extracellular vesicles (herein named BcEVs). In medical mycology, such explorations have become important during the last decade [42,45], but they are still in their infancy in phytopathology. EVs have only been described in the cotton pathogen *F. oxysporum* [59]; the cereal pathogens *F. graminearum*, *Zymoseptoria tritici*, and *Ustilago maydis* [60,94]; and the polyphagous *C. higginsianum* [61].

By using transmission electron microscopy (TEM), BcEVs of diverse shapes, sizes and electron densities can be observed between the cell wall and the plasma membrane of hyphae grown in liquid or solid media. Electron tomography further revealed that BcEVs co-exist as discrete vesicles and nanotubes, structures not yet reported in pathogenic fungi but observed in humans [95] and in symbiotic arbuscular mycorrhizal fungi [96,97]. BcEVs mainly appear in groups, and tomography data support their possible origin in the fusion of multivesicular bodies (MVB) with the plasma membrane. This corroborates previous observations in the human pathogen *C. neoformans* [98], including our parallel observation of single vesicles whose origin could be the budding of the plasma membrane. Lastly, BcEVs were also observed in hyphae interacting with plant cells, strengthening the biological meaningfulness of in vitro observations and corroborating observations in another phytopathogenic fungus [61].

BcEVs collected by differential centrifugations of liquid culture supernatants formed gelatinous pellets that were very difficult to manipulate. Considering the EVs contamination by glucurono-xylomannan when *C. neoformans* is grown in liquid medium [99], it is possible that a component of polysaccharidic nature sediments alongside the BcEVs when prepared this way. An improved protocol was established that relies on solid fungal cultures and an additional BcEVs purification step via a density gradient. This protocol has a very good reproducibility and aligns with the recommendations of the latest guidelines communicated by the International Society for Extracellular Vesicles MISEV2018 [18]. It yielded EVs of typical shapes and sizes, as attested by TEM and DLS analysis. In addition, the proteomic analysis of these BcEVs showed a satisfactory low (8.8%) amount of predicted SP-containing proteins. In our hands, as in *C. neoformans* [100], the preparation of BcEVs benefits from harvesting mycelia from solid cultures. In addition, their density-gradient purification reduces the risk of contamination by co-pelleting non-EV molecules.

A total of 2461 proteins were identified in BcEVs. According to the Fungal EV database 2021 (http://exve.icc.fiocruz.br, accessed on 15 December 2021). This is an intermediate situation between the 3312 and 3037 proteins found in *C. neoformans* and in *A. fumigatus* on the one hand, and the 1410 and 1256 proteins found in *Histoplasma capsulatum* and in *C. albicans* on the other hand. Based on the low variation in abundance between 4 biological replicates, 673 of these proteins were selected as the best candidates of BcEV proteins. The predicted distribution and functions of these proteins are very similar to those of the proteins identified in EVs produced by mammalian cells and fungi [42,43], including human and plant pathogenic species [41,59,61]. First, these proteins are cytosolic or TM and GPI-anchored proteins, two categories of EV proteins defined by Théry et al. [18]. Second, they are predicted to localize to many of the cell compartments, including the cell membrane, the cytoplasm, the mitochondria, the endoplasmic reticulum, the Golgi apparatus and the nucleus. Third, their functional classification highlights typical categories found in EVs, such as transport, metabolism, proteostasis, oxidoreduction, cell wall remodeling, vesicular trafficking, translation and cytoskeleton. Fourth, the BcEVs proteins include acknowledged EV markers such as HSP, ESCRT, SNARE, Sec, and G proteins. Notably, they also include ten orthologs of established EV markers in *C. albicans* [87] and two markers found in mammalian and plant EVs that have not yet been reported in fungal EVs: the autophagy-related protein Atg8 and one tetraspanin [11]. Altogether, these data suggest that BcEVs, as a population of vesicles, carry a complex mixture of proteins (Figure 8). How this diversity of proteins is loaded into the BcEVs is unclear, but our dataset highlights several putative protein interaction networks; therefore, it is possible that protein complexes are taken up when the vesicles are formed.

It is established that fungal EVs can transfer a range of bioactive molecules to target cells, serving the purpose of cell communication or modulating interactions with hosts [16,28,40,101,102,103]. Here, we present evidence that fluorescent BcEVs can label the cell wall of hyphae from *B. cinerea* and *F. graminearum,* while they do not label cells of *S. cerevisiae*. Furthermore, the fluorescent signal can later be detected inside cells of *B. cinerea* and *F. graminearum*. This could suggest a role for these vesicles in intra-species communication as well as in specific inter-species communication with a subset of fungi present in the microbiomes where *B. cinerea* lives, as a saprobe or as a pathogen. EV-mediated communication in fungi requires outwards crossing of the cell wall of the donor cell, interaction with the wall of the recipient cells and inwards crossing of the latter. These processes are poorly understood, but they must involve surface components of the vesicles. In this study, 271 membrane and GPI-anchored proteins were identified in BcEVs, among which 16 proteins involved in fungal cell wall biogenesis. Such proteins, identified in other fungal EVs and proposed to play a role in FCW remodeling, could participate in the crossing of FCW by BcEVs [28,31,41,61,104,105]. Furthermore, BcEVs contain cytoskeleton proteins, and these proteins are postulated to play a role in the passage of EVs across cell walls [106]. Lastly, several metabolic enzymes, such as transaldolases, aldolases, glyceraldehyde-3-phosphate dehydrogenase, alcohol dehydrogenase and a phosphoglycerate kinase are among the proteins identified in BcEVs. Interestingly, all these enzymes were detected in EVs of human pathogenic fungi and could play a role in their adhesion to host cells [104].

This study demonstrates that fluorescent BcEVs can stain onion cells. Mirroring the previously described transfer of material from a host plant to *B. cinerea* [56], this may suggest that BcEVs can transfer fungal material to plant cells. Related to this, it is notable that BcEVs contain the necrosis inducer BcGs1, the virulence factor BcSun1 and several lytic enzymes (i.e., proteases, laccases, esterases) similar to those found in EVs of other pathogenic fungi [49,107]. Predicted apoplastic proteins and few predicted effectors were also identified in the BcEV proteome. However, under the conditions used to produce BcEVs, the latter showed no toxic effect on bean or tobacco leaves. Further research is needed to investigate the factors that could influence their composition and putative action on plant cells [103,108,109], including fungal cultures in the presence of or in contact with plant material.

The impact of BcEVs on recipient cells can derive from either, or both, surface and lumenal compounds. For instance, the membrane chitin and glucan synthases can rescue yeast cells from antifungal molecules [28], while proteins related to glucose, phosphate or iron utilization can improve nutrition [42]. Other unidentified components can act as stress-message effectors that modify or boost fungal growth [40,101]. Here, we report a positive effect of BcEVs on the growth of *B. cinerea* that is not observed with *F. graminearum*. The multiple membrane transporters identified in BcEVs (including phosphate, sugars, amino-acids and iron carriers), as well as cell-wall-related proteins or the sugars, lipids/sterol, amino-acids and phosphate-related metabolic enzymes could stimulate the growth of *B. cinerea* cells. The same applies to the ribosomal proteins, to proteins involved in traffic or proteostasis, or to enzymes of the respiratory chain. However, the reason why these proteins would not operate in *F. graminearum* cells is unclear.

## 5. Conclusions

In conclusion, this study documents the production and isolation of EVs by *B. cinerea*. This multidisciplinary approach revealed complementary biological, biochemical and functional information. These EVs are heterogeneous in shapes, sizes and electron densities. Similar to EVs from human fungal pathogens, they are carriers of both membrane and soluble intracellular and extracellular proteins involved in diverse biological processes. Their potential role in intra and interspecies communication has been revealed. How this communication affects the recipient cells remains to be elucidated.

## Figures and Tables

**Figure 1 jof-09-00495-f001:**
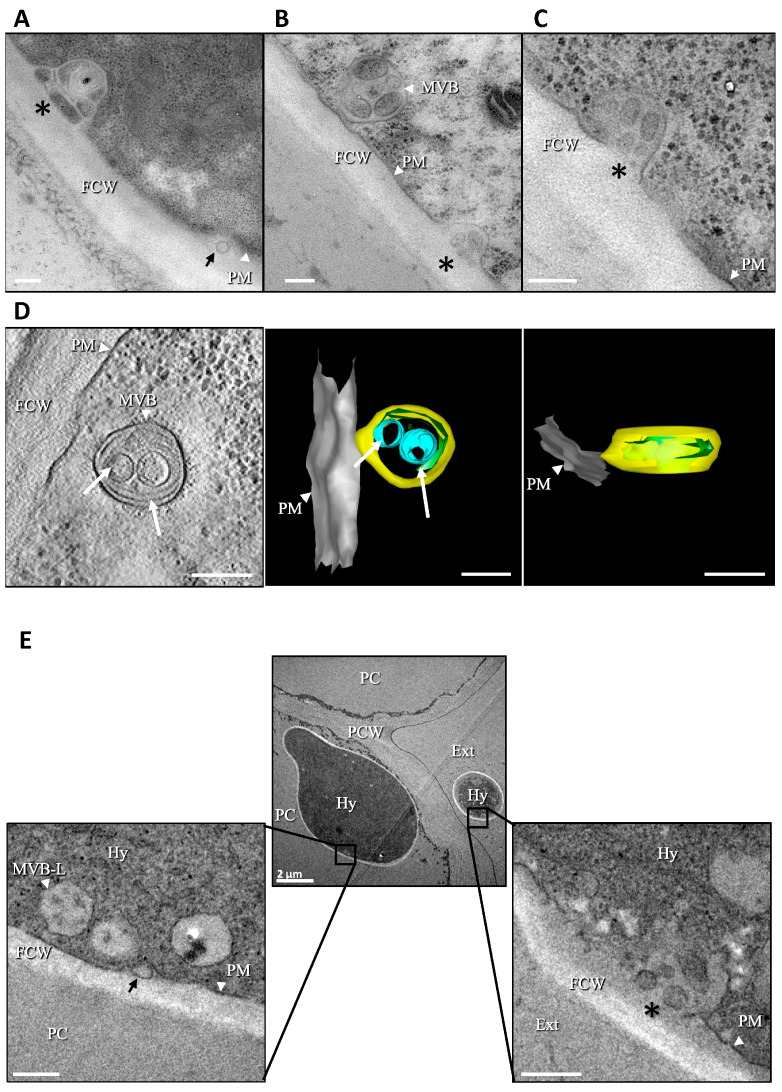
**Hyphae of *B. cinerea* produce EVs in vitro and *in planta***. (**A**–**C**) TEM of *B. cinerea* hyphae grown in vitro. Single (black arrow) and groups (black star) of extracellular vesicles (BcEVs) are visible between the plasma membrane (PM) and the fungal cell wall (FCW). Multi-vesicular bodies are indicated (MVB). Images are representative of hyphae collected from either solid or liquid cultures. (**D**) Electron tomography of a MVB adjacent to the plasma membrane (hypothesized pre-fusion stage) and containing ovoid (blue) and tubular (green) BcEVs. (**E**) TEM images of *B. cinerea* hyphae (Hy) located on the external aerial side of bean epidermis cells (Ext) and inside plant cell cytoplasm (PC). The plant cell wall is indicated (PCW, made with parallel lines), covered by the cuticle in the outer-periclinal sides of the two epidermis cells. At higher magnification, single (black arrow) and groups (black star) of BcEVs are visible between the fungal PM and the FCW (more translucent material) as well as MVB-like structures (MVB-L). Scale bar: 200 nm unless indicated.

**Figure 2 jof-09-00495-f002:**
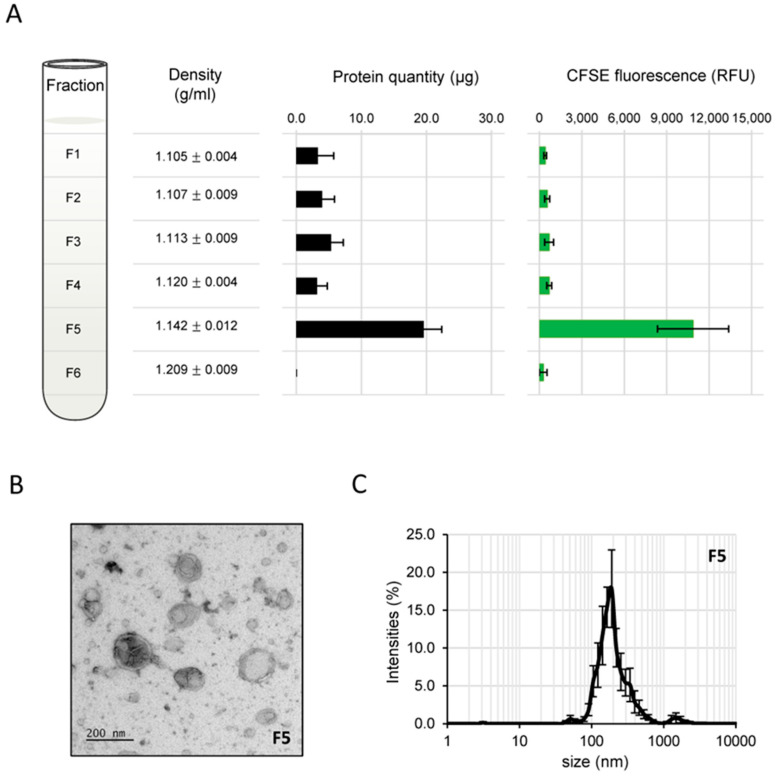
**Isolation of EVs from solid cultures of *B. cinerea*.** (**A**) EVs were isolated using differential centrifugation and further separation by density (iodixanol) gradient ultracentrifugation. The fractions (F1 to F6) were analyzed through measurements of their densities (9 biological replicates), protein concentrations (28 biological replicates, F5: 19.6 ± 4.7 µg/4.6 ± 0.1 g dry mycelium) and labelling with fluorescent CFSE (6 biological replicates). Protein and fluorescence quantifications were performed after washing of the iodixanol and collection of the biological material by centrifugation. Results are expressed as mean values ± SEM. (**B**) TEM of fraction F5, showing vesicles of different sizes and densities. (**C**) Dynamics light scattering of fraction F5, showing particles mostly (>95%) in the range of 90 to 500 nm in size (8 biological replicates).

**Figure 3 jof-09-00495-f003:**
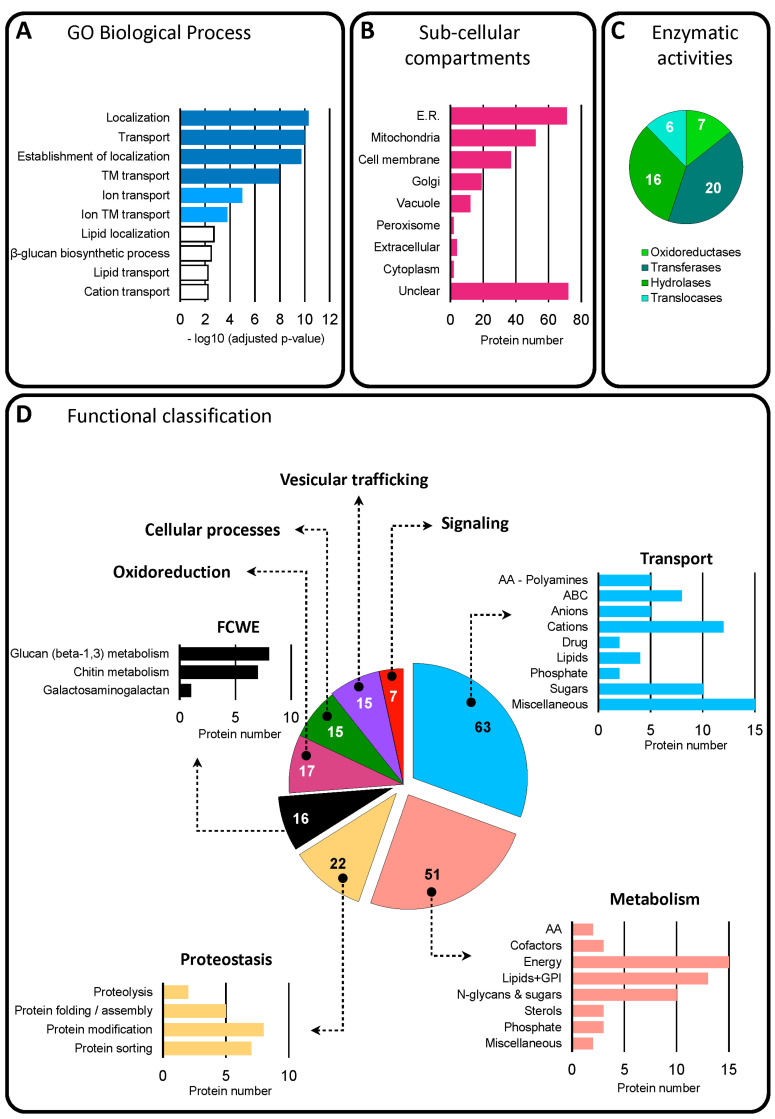
**Diversity of the 271 membrane and GPI-anchored proteins identified in BcEVs**. (**A**) Top 10 enriched biological processes (dark blue, *p*-value < $10^{-8}$; blue, *p*-value < $10^{-7}$; white, $10^{-6}$ < *p*-value < $10^{-3}$). (**B**) Predicted sub-cellular compartments. (**C**) Predicted enzymes. (**D**) Classification of the 206 proteins with a predicted function into 8 categories and 24 sub-categories. The proteins with unknown functions are not represented. The number of proteins inside each pie chart is indicated in white or black.

**Figure 4 jof-09-00495-f004:**
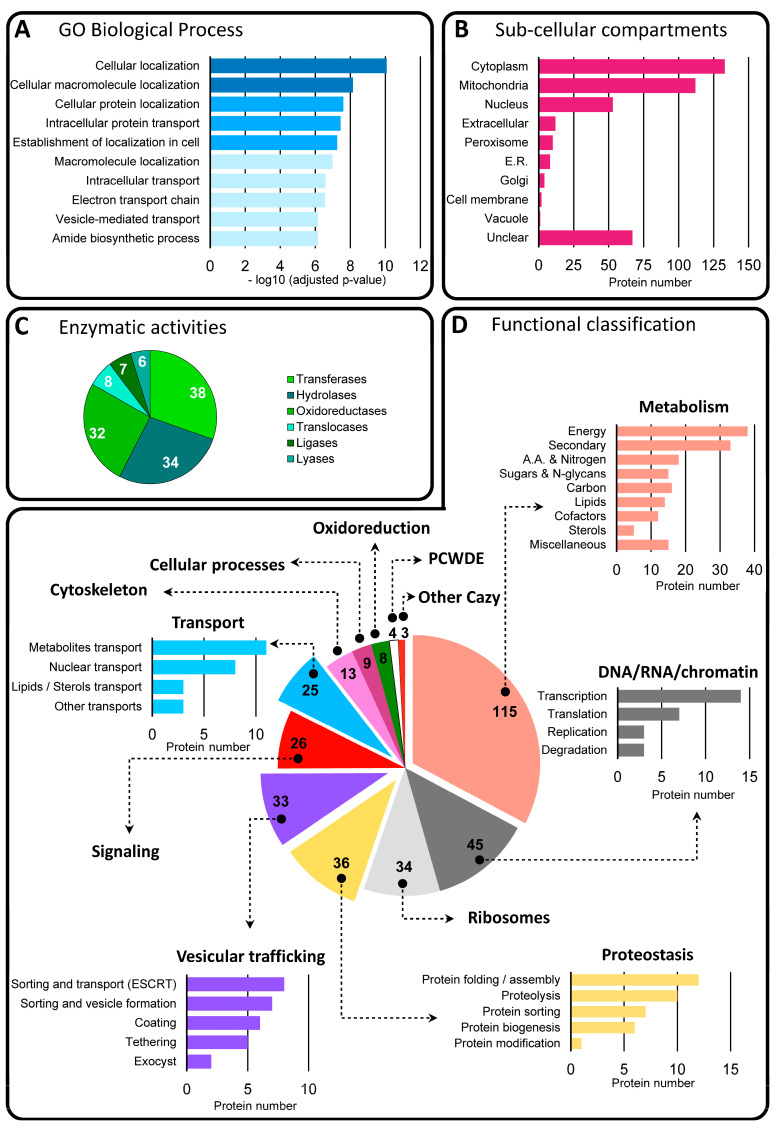
**Diversity of the 402 soluble proteins identified in BcEVs.** (**A**) Top 10 enriched biological processes (dark blue, *p*-value <$10^{-8}$; blue, *p*-value <$10^{-7}$; light blue, *p*-value <$10^{-6}$). (**B**) Predicted sub-cellular compartments. (**C**) Predicted enzymes. (**D**) Classification of 351 proteins with a predicted function into 12 categories and 27 sub-categories. The proteins with unknown functions are not represented. The number of proteins inside each pie chart is indicated in white or black.

**Figure 5 jof-09-00495-f005:**
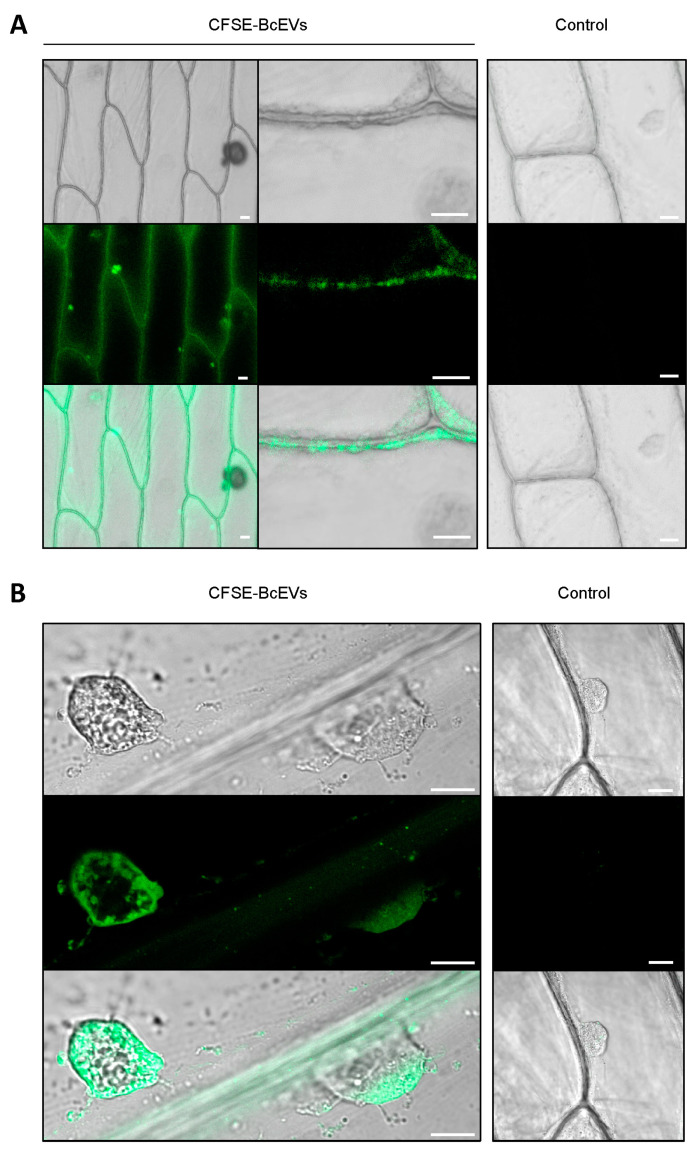
**BcEVs stain onion cells**. BcEVs labelled with CFSE were deposited onto onion epidermal cells, incubated for 3 h and washed away before imaging by confocal microscopy. (**A**) Low magnification of the CFSE labelling of the epidermal cells (**left**) and higher magnification showing the labelling of the plasma membrane (**center**). (**B**) Subcortical nuclear zones of adjacent epidermal cells showing intracellular CFSE labelling. The absence of fluorescence was verified in the epidermal cells exposed to the PBS control sample (see Section 2.3). Scale bar: 20 µm.

**Figure 6 jof-09-00495-f006:**
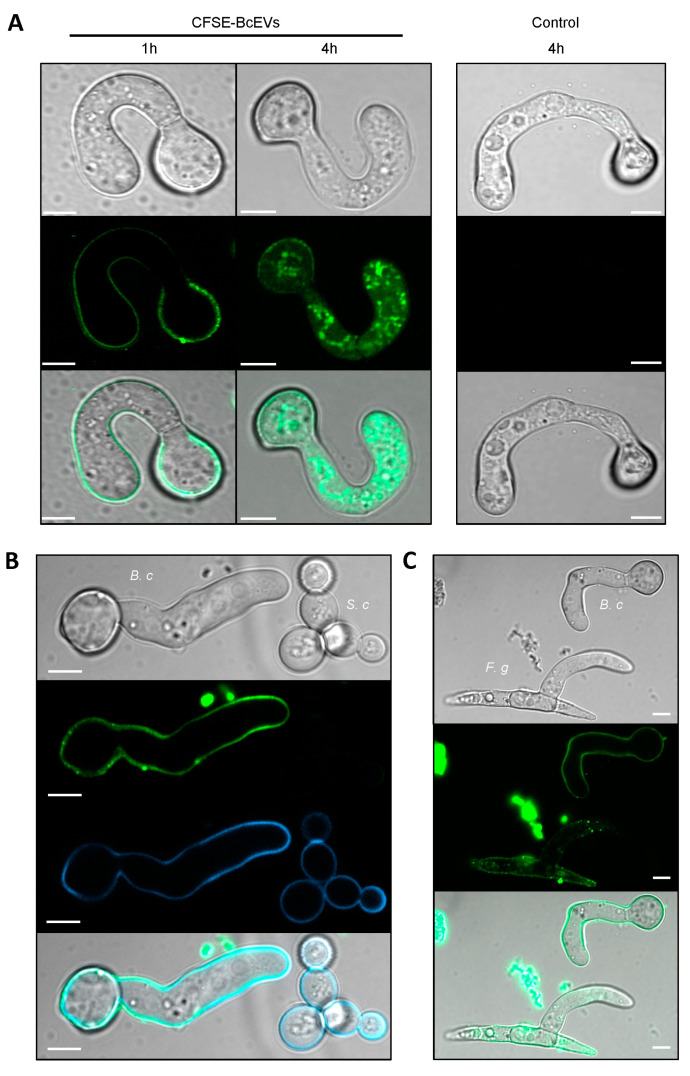
**BcEVs stain cells of *B. cinerea* and *F. graminearum*, but not that of the yeast *S. cerevisiae*.** (**A**) Confocal microscopy of germinated *B. cinerea* conidia (*B. c*) incubated for 1 h and 4 h with CFSE-BcEVs. The absence of green fluorescence (CFSE) was verified in hyphae exposed to the PBS control sample (control). (**B**) Yeast cells (*S. cerevisiae*) collected in exponential phase and incubated for 1 h with CFSE-BcEVs. Co-staining with aniline blue (glucan marker) is shown. (**C**) Germinated conidia of *F. graminearum* (*F. g*) incubated for 1h with CFSE-BcEVs. Fluorescent BcEVs aggregates are visible in B and C (strong green fluorescent signal outside the fungal cells). Scale bar: 5 µm.

**Figure 7 jof-09-00495-f007:**
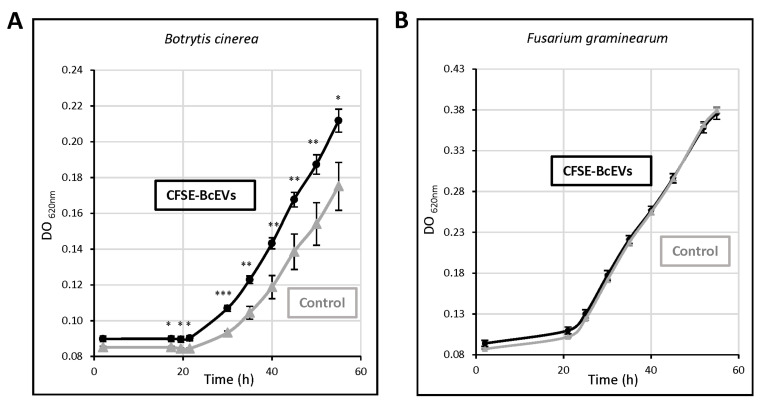
**BcEVs can improve the growth of *B. cinerea*.** (**A**) Growth curves of *B. cinerea* in liquid medium supplemented with CFSE-BcEVs or with PBS control sample. (**B**) Growth curves of *F. graminearum* grown under the same conditions used for *B. cinerea* in (**A**). Six biological replicates were performed. Data are expressed as mean values ± SEM (Student’s *t*-test *p*-value * <0.05; ** <0.01; *** <0.001).

**Figure 8 jof-09-00495-f008:**
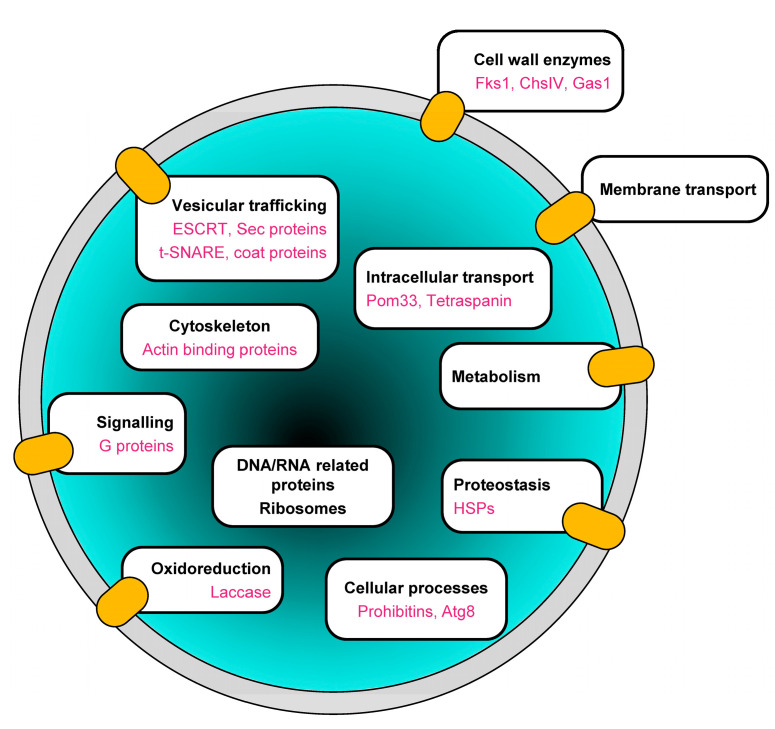
**Schematic representation of BcEVs.** The diverse functions highlighted by the BcEVs proteomic analysis are indicated, with orange ovals positioned where membrane proteins have been identified. EV marker proteins are indicated in pink.

## Data Availability

The mass spectrometry proteomics data were deposited to the ProteomeXchange Consortium via the PRIDE partner repository with the dataset identifier PXD035766.

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
