# Peer review of "Extracellular Vesicles of the Plant Pathogen Botrytis cinerea"

_jof, 2023, doi:10.3390/jof9040495_

Round 1

Reviewer 1 Report

Routes for non-conventional (no signal peptide) secretion are important and recent work implicates extracellular vesicles (EV) in fungal-host communication and virulence. Here, the EVs of a generalist, necrotrophic plant pathogen were studied for the first time. Formation of EVs and their interaction with fungal and plant partners were characterized. Methods to enrich for EVs were optimized (purification from solid culture).

Proteomics of the EV fraction showed that the overall number of proteins and functional categories are similar to those for the other fungi in the literature. In this part, the study seems to fall short of expectations somewhat, in that the proteomic data set might have been used further to compare with existing data from other pathogens. Do the EVs contain predicted effectors? Maybe I've somehow missed it, as effector prediction does appear in the Methods section. The repertoire of Botrytis cinerea effectors is not so well characterized, yet there are several papers on effectors; combined with the tools available for prediction, and the genome sequence, some more effort in this direction might raise the significance of this paper.

minor notes:

line 25 in the abstract - "a few" might be more appropriate as there are several papers, indeed well-cited in the manuscript. 

line 68 "it is admitted" better: "it is thought that" or "it has been proposed that" or "it has been shown in some cases that" - depending on the meaning intended.

line 71 define MVB here (first use)

line 295 "whose putative origin could not be captured" - not clear; the meaning does become clear by line 301 - the idea of "captured" is that there are images with pre- and post fusion events: maybe discuss the larger population first, and then say that second one had unclear origin. 

Table 2 - typo organel => organelle

Figure S3 - please define in the legend - STRING color coding

Author Response

As suggested, the proteome analysis of BcEVs was completed by a search of effectors, starting from the few known in B. cinerea and then exploring all the effectors known in phytopathogenic fungi and used by K Seong & V Krasileva to cluster putative effectors according to their protein structure (nature microbiology, 2023). The presence of one known B. cinerea virulence factor in the proteomic dataset is also now mentioned in the result section, as well as the predicted apoplastic proteins.

Minor notes:

Corrections or modifications were made to take all minor comments into consideration.

Author Response

1/ The resolution of Figure 4 was increased.

2/ A schematic representation of a BcEV, including the marker proteins, was inserted in the discussion session. It was not presented as the representation of a typical BcEV because microscopy showed that BcEVs constitute a population of multiple vesicles of different sizes and shapes, the individual composition of which is not known.

3/ In figure 7, the stimulation of B. cinerea growth by BcEVs is shown and a negative control is included in the experiment (absence of EVs). In parallel, the demonstration that BcEVs do not stimulate the growth of another filamentous fungus, F. graminearum, is made. In the case of both species, fungal growth is measured by absorbance.

The use of a fungistatic molecule in this experiment would evidently block fungal growth and result in no increase in absorbance. If the EVs had inhibited the growth of B. cinerea, we could see the point of using such a control experiment, but we do not understand how this particular experiment would represent a negative control of the observed positive effect of EVs. If there would be a point that we missed in your proposition of using a fungistatic molecule, please let us know, but otherwise we believe that the negative control we used is more appropriate and we therefore wish to keep this part of the result section as it is.